# Systematic Review on Influenza Burden in Emerging Markets in 2018–2023—An Evidence Update to Guide Influenza Vaccination Recommendations

**DOI:** 10.3390/vaccines12111251

**Published:** 2024-11-02

**Authors:** Moe H. Kyaw, Sophie Bozhi Chen, Shishi Wu, Chee Yoong Foo, Verna Welch, Constantina Boikos, Oladayo Jagun

**Affiliations:** 1Pfizer Inc., Collegeville, PA 19426, USA; verna.welch@pfizer.com (V.W.); constantina.boikos@pfizer.com (C.B.); 2Real World Solutions, IQVIA, Singapore 079906, Singapore; sophiebozhi.chen@iqvia.com (S.B.C.); shishi.wu@iqvia.com (S.W.); cheeyoong.foo@iqvia.com (C.Y.F.); 3IQVIA, Parsippany, NJ 07054-2957, USA; oladayo.jagun@iqvia.com

**Keywords:** influenza, disease burden, emerging markets

## Abstract

**Background:** Influenza is a contagious respiratory illness responsible for seasonal epidemics and with potential to cause pandemics. The decline in influenza-related studies published since 2018 resulted in data gaps, particularly in emerging markets. **Methods:** This systematic review searched for studies in six databases and gray literature sources to define the clinical burden of influenza and influenza-like illness (ILIs) and their associated sequelae among humans across emerging markets. Eligible studies were published in English, Spanish, or Chinese between January 2018 and September 2023 and conducted in Asia, the Middle East, Africa, and Latin America. **Results:** In total, 256 articles were included, mostly on lab-confirmed influenza infections (n = 218). Incidences of lab-confirmed influenza cases in Asia (range 540–1279 cases/100,000 persons) and Sub-Saharan Africa (range 34,100–47,800 cases/100,000 persons) were higher compared to Latin America (range 0.7–112 cases/100,000 persons) and the Middle East and North Africa (range 0.1–10 cases/100,000 persons). Proportions of lab-confirmed influenza cases and influenza-associated outcomes (i.e., hospitalization, ICU admission and death) varied widely across regions. Temporal variation in influenza trend was observed before and during the COVID-19 pandemic. **Conclusions:** In conclusion, influenza causes significant disease burden in emerging markets. Robust large real-world studies using a similar methodology are needed to have more accurate estimates and compare studies within age groups and regions. Continuous monitoring of influenza epidemiology is important to inform vaccine programs in emerging markets with heavy influenza disease burden.

## 1. Introduction

Influenza is a contagious respiratory illness which contributes significantly to the global disease burden. According to the WHO’s estimation, approximately 3 to 5 million severe influenza cases and 290,000 to 650,000 influenza-associated deaths occur annually [1]. The regions of North America, Europe, East Asia, and South Africa have higher influenza activity during the winter [2,3]. In tropical and subtropical regions, the influenza pattern is less predictable and can occur year-round, often with multiple peaks [4]. Regions such as Southeast Asia and North Africa experience a more constant presence of influenza, with possibly annual varying intensities [3,4,5]. The majority of influenza infections are self-limiting and do not require medical attention. A subset of influenza infections can result in severe complications and sequelae, which increase the risk of hospitalization, intensive care unit (ICU) admissions, and death [6,7,8,9,10]. Sequelae are classified as acute or long-term sequelae. Acute sequelae may include complications of the respiratory tract (e.g., pneumonia), heart (e.g., myocarditis), central nervous system (e.g., Guillain–Barré syndrome), kidneys (e.g., acute renal failure), and other organs [11]. Long-term sequelae may lead to reduced lung function, reduced quality of life, functional decline in older adults due to prolonged bed rest, and worsening of diabetes-related complications [8,12,13].

Influenza is caused by influenza viruses, which are enveloped viruses with a segmented RNA genome and categorized into four main types: A, B, C and D [14,15]. Influenza type A and B are most prevalent in humans and responsible for seasonal flu epidemics [16,17]. Due to its high morbidity and mortality, influenza type A has potential for causing pandemics and poses a significant public health concern [18]. Influenza type B and C cause mild respiratory illness, though type C is less common and primarily affects children and adolescents without causing widespread outbreaks [19,20]. Influenza type D has been recently discovered, mainly infects cattle, and remains unrecorded in humans [14].

Influenza can be diagnosed by clinical diagnosis of influenza-like illness (ILI) and/or by laboratory confirmation of influenza infection [21]. Most common laboratory tests include molecular assays (e.g., reverse transcription polymerase chain reaction (RT-PCR)), which offer highly accurate results by detecting the genetic material of the influenza virus and antigen detection tests (e.g., rapid influenza diagnostic tests), which provide rapid results but may be less sensitive [21]. Less frequently used is viral culture, where the influenza virus is grown in laboratories [21]. In settings with limited laboratory testing capacity, healthcare providers diagnose influenza based on clinical symptoms and patient history (i.e., ILI). The World Health Organization (WHO) defines ILI as “an acute respiratory illness with a measured temperature of ≥38 °C and cough, with onset within the past 10 days” [22]. The clinical diagnosis of ILI has limited sensitivity and specificity in detecting PCR-positive influenza, due to similarity in symptoms among various respiratory viruses, including COVID-19. Less than 25% of ILI cases yields a positive RT-PCR test [23]. Consequently, the diagnosis of ILI may overestimate influenza’s disease burden. The WHO recommends the use of laboratory-confirmed outcomes whenever possible [24].

The influenza pandemic in 2009 prioritized implementing preventive measures for influenza in public health policies. It became apparent that a standardized approach to epidemiological surveillance for influenza was needed, and in response, the WHO developed global epidemiological surveillance standards for influenza [25]. However, particularly in emerging markets, influenza surveillance is still limited [26]. Seasonal influenza vaccination is an effective preventive measure, though a recent meta-analysis showed low global influenza vaccination rates: 25% in the general population, 42% in persons with chronic diseases, 37% in healthcare workers, and 26% in pregnant women [27]. Geographic differences were highlighted, with relatively high influenza vaccination rates in the American region, low rates in the European and Western Pacific regions, and even lower rates in the Eastern Mediterranean, Southeast Asian, and African regions [27].

Existing systematic reviews on global influenza disease burden overlook cross-regional nuances [28,29,30,31] and reviews aimed at specific countries provide limited evidence for direct comparisons of the influenza burden across different geographical regions [5,32,33]. This absence of cross-regional comparisons of influenza disease burden limits our understanding of the global variability in influenza impact, particularly in emerging markets, which hinders the development of targeted and effective public health strategies (e.g., vaccine programs) that cater to the specific needs of different regions.

The objectives of this systematic review were to determine the clinical burden of influenza and ILI infections and their associated acute and long-term sequelae in emerging markets from January 2018 until September 2023. Additionally, the impact of the COVID-19 pandemic on the influenza disease burden was explored. Emerging markets include countries in Asia (excluding China, Hong Kong, Macau, and Japan and including Taiwan), Africa, the Middle East, and Latin America.

## 2. Materials and Methods

The systematic review was conducted and reported according to a protocol published in the PROSPERO database (registration ID: CRD42023457576). Reporting of the systematic review adhered to the Preferred Reporting Items for Systematic reviews and Meta-Analyses (PRISMA) guidelines [34].

### 2.1. Databases and Search Strategy

Systematic literature searches were conducted on 8 September 2023 in three predominantly English-language databases: PubMed, Embase, and Cochrane Library. Additionally, Spanish-language articles were searched for in two non-English databases, the Latin American and Caribbean Health Sciences Literature (LILACS) and LA Referencia (Federated Network of Institutional Repositories of Scientific Publications), and Chinese-language articles in the Airiti Library. The search strategy was built with controlled vocabularies, e.g., Emtree terms for Embase, and free text searches encompassing terms relevant to influenza, ILI, its sequelae, and the outcomes of interest (Appendix A). Search limits were applied for language (English, Spanish, and Chinese), publication period (articles published after 1 January 2018), and geographical regions (Asia excluding China, Hong Kong, Macau, and Japan and including Taiwan; Africa; the Middle East; and Latin America).

The electronic database searches were supplemented with gray literature searches on 19–20 September 2023 in Google Scholar, the WHO Global Influenza Program, Global Influenza Hospital Surveillance Network (GIHSN), and country-level influenza surveillance outputs from national agencies such as the Ministry of Health. The key words “influenza infection”, “influenza like illness”, “influenza sequalae”, and “influenza complication” were used to search for the literature. Only countries where English, Spanish, or Chinese are the official languages or countries with surveillance outputs available in English were included in these searches.

### 2.2. Screening

The title/abstract and full-text screening were executed by two reviewers. Initially, both reviewers independently processed 10% of the studies. The inter-reviewer agreement was evaluated using the kappa coefficient, with coefficients exceeding 0.75 indicating good agreement between reviewers [35,36]. Any discrepancies were resolved by a third reviewer. Subsequently, the primary reviewer proceeded with the remaining 90% of studies.

After deduplication, the titles and abstracts of the retrieved records were screened against prespecified eligibility criteria (Appendix A). These criteria were developed using guidance from the Population, Exposure, Comparator and Outcome (PECO) framework [37]. Interventional, observational, modeling, and surveillance studies with data on the clinical burden of influenza and ILI infection and their associated acute and long-term sequelae among human populations across emerging markets were included. Sequelae were limited to influenza- or ILI-associated pulmonary and extra-pulmonary complications that have been clinically diagnosed or indicated using clinical diagnostic codes, such as the International Statistical Classification of Diseases and Related Health Problems (Appendix A). Articles identified as potentially relevant based on the title/abstract screening were retained for full-text assessment. Moreover, reference lists of included studies and selected systematic reviews and meta-analyses were checked. The screening procedures were conducted using the web-based literature review software DistillerSR (DistillerSR Inc., Ottawa, ON, Canada, 2023).

### 2.3. Data Extraction and Quality Appraisal

Similarly to the screening process, two reviewers independently executed the data extraction and quality appraisal of the initial 10% of included studies; the remaining 90% was processed by one reviewer. Information was extracted on the influenza variant, influenza category, and burden outcomes associated with influenza, ILI, influenza sequalae, and ILI sequalae (Appendix A). In addition, information related to sample and design was extracted, including study period, study design, setting, data source, population group, sample size, participants’ enrollment period, study follow-up period, statistical analysis methods, factors adjusted in the statistical model, sex distribution, mean age, vaccinated population, and other comorbidities. A standardized Microsoft Excel^®^ form was used for extracting data from included studies.

The study quality was assessed with the QualSyst tool based on 8 criteria (study design and research question, definition of outcomes and exposures, reporting of bias and confounding, and sufficient reporting of results and limitations) to calculate a summary score for each study [38]. Using a pre-designed Microsoft Excel^®^ form, each criterion was scored as 0 (unfulfilled or inapplicable criteria), 1 (partially fulfilled criteria), or 2 (fully fulfilled criteria). The studies were further grouped into 3 categories based on their quality scores: low quality (score < 50%), medium quality (score 50–70%), or high quality (score ≥ 70%) [39,40]. Data extracted directly from surveillance databases were not subjected to quality assessment.

### 2.4. Data Synthesis

The data were analyzed narratively in tables and figures and stratified by region: Asia, Sub-Saharan Africa, the Middle East and North Africa, and Latin America. All studies that met the eligibility criteria were included in the data synthesis. Studies were further tabulated by subgroups of interest, including outcome types, sequalae types (where applicable), and study period (before or during COVID-19).

Lab-confirmed influenza infection was defined as diagnosed by RT-PCR or antigen detection tests and ILI as diagnosed by clinical diagnosis. Outcomes of interest were incidence and proportion of influenza and ILI cases, hospitalizations, ICU admissions, mortality, outpatient visits, and emergency room (ER) visits. A similar approach was employed for influenza- and ILI-associated complications. Given the absence of consensus in the literature regarding the definitions of acute and long-term influenza-associated sequelae, a defined framework was adopted based on the timeline of complication identification. Complications recorded within 30 days from the influenza diagnosis were classified as acute and between 30 and up to 90 days after as long-term [41,42,43]. The time period was defined as pre-COVID-19 (2018–2019) and during COVID-19 (2020–2023). In the Results section, incidence rates are reported as ranges and proportions as medians with interquartile range (IQR). Evidence gaps were systematically summarized.

## 3. Results

Through six databases 3905 unique records were identified; of these, 3220 records were excluded during title/abstract screening and 450 articles during full-text screening. The reasons for exclusion at full-text screening are enclosed in Figure 1. In the gray literature, 602 records were identified, of which 581 reports were excluded. In total, 256 articles were included for data synthesis. The full reference list is enclosed in the Appendix A.

### 3.1. Study Characteristics

The characteristics of the included studies are summarized in Table 1. Most studies were conducted in Asia (n = 119), followed by Latin America (n = 56), the Middle East and North Africa (n = 47), and Sub-Saharan Africa (n = 45). Cross-sectional studies accounted for 45% of the study designs, cohort studies 33%, and surveillance studies 19%. The population comprised both children (<18 years old) and adults (≥18 years old) in 44% of the studies. In almost 30% of the studies, no age range was reported. Vaccinated populations were included in 25% of the studies and the proportion of vaccinated study participants ranged from 0.1% to 98%. Influenza type A (mostly H1N1 and H3N2 subtypes) and influenza type B (lineage mostly not specified) were studied in 70% and 57% of the studies, respectively. Most evidence focused on lab-confirmed influenza infection (n = 218) and its sequelae (n = 72), with limited evidence on ILI (n = 45) and its associated sequelae (n = 0). Lab-confirmed influenza was most frequently studied in South Korea (Asia), Zambia (Sub-Saharan Africa), Brazil (Latin America), and Egypt (Middle East and North Africa). For ILI, most studies were conducted in India (Asia), Zambia (Sub-Saharan Africa), Colombia and Peru (Latin America), and Egypt (Middle East and North Africa). A complete overview of the available evidence is shown in Appendix A. Most studies had a high quality based on the QualSyst tool: 91% of the studies on lab-confirmed influenza, 82% of the studies reporting lab-confirmed influenza-associated sequelae, and 71% of the ILI studies (Appendix A).

### 3.2. Disease Burden of Lab-Confirmed Influenza Infection

Incidence rates of lab-confirmed influenza varied widely, with data not uniformly available across regions, scarce data for ICU admission and outpatient visits, and missing data for ER visits (Figure 2, Appendix A). Temporal variation in influenza trend was observed before and during the COVID-19 pandemic. Incidences of influenza cases were higher in Asia (range 540–1279 cases/100,000 persons) and Sub-Saharan Africa (range 34,100–47,800 cases/100,000 persons) compared to Latin America (range 0.7–112 cases/100,000 persons) and the Middle East and North Africa (range 0.1–10 cases/100,000 persons). Evidence from Latin America and the Middle East showed a trend of decreasing incidence rates from 2019 to 2022. Studies in Asia reported higher incidence rates of lab-confirmed influenza hospitalization compared to the other regions, with studies indicating rates up to 9590 hospitalizations/100,000 persons, and higher rates observed in children and adolescents than adults. The highest incidence of influenza-associated death was reported in Sub-Saharan Africa (range 53–247 deaths/100,000 persons). In Asia, the incidence rate of influenza deaths decreased over the years 2018–2020.

Proportions of lab-confirmed influenza cases and influenza-associated outcomes also varied widely across the regions (Figure 3, Appendix A). Temporal variation in influenza trend was observed before and during the COVID-19 pandemic. Sub-Saharan Africa reported the highest median proportion of influenza cases of 12.8% (IQR 14.0%). Influenza type A, particularly H1N1 and H2N3, contributed to most cases across regions. Influenza hospitalization proportions showed fluctuations over the years, with some regions experiencing decreasing trends (e.g., Latin America) and others without consistent trend (e.g., Asia). Children in Asia generally exhibited lower hospitalization proportions (0–41%) compared to adults (0–85%). In Asia and Sub-Saharan Africa, H1N1 and H2N3 contributed to the most hospitalizations; the Middle East and North Africa and Latin America also reported H3N2. The proportion of influenza-associated ICU admission was highest in Latin America with a median of 23.0% (IQR 28.0%). In Asia and Latin America, the ICU admission proportion showed a wider and higher range in adults compared to children and adolescents, and influenza type A contributed most. Studies in Middle East and North Africa also reported a high proportion of ICU admission attributed to influenza type B. The highest median proportion (6.6%; IQR 13.8%) of influenza-associated death was observed in the Middle East and North Africa. Trends over time varied between the regions. Latin America reported a higher proportion associated with H3N2 and H1N1, while the Middle East had an equal distribution of death between influenza type A (H3) and type B (unspecified). The median proportions of influenza-associated outpatient visits ranged between 9.7% (IQR 9.0%) and 12.7% (IQR 17.9%), with Asia reporting more H2N3 cases, Sub-Saharan Africa more H1N1, and Latin America a higher proportion of influenza type A than type B. Data on influenza-associated ER visits were very scarce.

### 3.3. Disease Burden of Lab-Confirmed Influenza-Associated Sequelae

Since only one study reported the incidence rate for influenza-associated long-term sequelae, Figure 4 provides an overview of the incidence rates related to acute sequelae. Limited and varying data were available on the outcomes of sequelae cases, hospitalization, death, and outpatient visits related to influenza-associated acute sequelae. More detailed information is reported in Appendix A.

The median proportions of influenza-associated acute sequelae are summarized in Figure 5, with more details provided in Appendix A. Asia had the most abundant data followed by Sub-Saharan Africa and the Middle East and North Africa. The highest median proportion of influenza-associated acute sequelae cases was reported for Sub-Saharan Africa (32.0%; IQR 79.6%), followed by Asia (10.3%; IQR 13.6%) and Latin America (0.3%; IQR 0.1%). The median proportion of hospitalizations due to acute influenza-associated sequelae ranged between 6.1% (IQR 2.9%) in Latin America to 21.3% (IQR 15.9%) in the Middle East and North Africa. The proportion of ICU admissions due to acute sequelae was highest in Asia with a median of 39.4% (IQR 42.1%) and mostly attributed to H1N1 and unspecified variants of type A. The median proportion of deaths due to acute influenza-associated sequelae in Asia was 13.0% (IQR 50.1%), with higher proportions reported in adults compared to children and adolescents. The limited data on outpatient visits attributed to influenza-associated sequelae exhibited significant variability without clear patterns. No studies reported proportions of ER visits due to influenza-associated sequelae.

### 3.4. Disease Burden of Influenza-like Illness

Data on incidence rate was only available for ILI cases and not for all regions (Appendix A). Asia reported a higher incidence rate (range 4980–11,690 cases/100,000 persons) than Sub-Saharan Africa (range 90–1030 cases/100,000 persons). Temporal trends indicate a decline from 2018 to 2019 in Asia’s incidence rate, in contrast to an upward trend observed in Sub-Saharan Africa.

Across the regions, proportion data were available for ILI cases, hospitalization, death, and outpatient visits (Appendix A). The median proportion of ILI cases ranged from 9.9% (IQR 14.1%) in Asia to 19.3% (IQR 56.0%) in the Middle East and North Africa. These proportions varied by year and age group, showing fluctuations without a consistent pattern emerging across regions. Sub-Saharan Africa had the highest proportion of ILI-associated hospitalizations with a median of 9.6% (IQR 8.1%) compared with Asia (3.7%; one study) and Latin America (median 0.3%; IQR 0.3%), though these proportions decreased in Sub-Saharan Africa from 2019 to 2020. The median proportions of ILI-associated deaths ranged from 0% (IQR 11.3%) in the Middle East and North Africa to 0.9% (IQR 5.9%) in Sub-Saharan Africa. A decrease in the proportion of deaths was observed from 2019 to 2020 in Sub-Saharan Africa. Proportions of ILI-associated outpatient visits varied widely across the regions; both Sub-Saharan Africa and Asia experienced a decrease in the proportion of outpatient visits from 2019 to 2020.

## 4. Discussion

### 4.1. Disease Burden of Lab-Confirmed Influenza

The data availability on outcomes related to lab-confirmed influenza varies among emerging markets. Asia and Sub-Saharan Africa had relatively more data on the incidence rates of influenza cases and hospitalizations. Information on ICU admissions and ER visits was scarce across all regions examined. Notably, we observed an increase in influenza-related studies in Sub-Saharan Africa since 2018, in contrast to earlier reviews that highlighted severe data gaps in this region. This may be a result of substantial improvement in surveillance capacity in the past decade [5].

Asia and Sub-Saharan Africa experienced the highest burden from influenza infections and severe outcomes: Asia reported the highest incidence rates of influenza-associated hospitalization, while Sub-Saharan Africa showed the highest rates of incidence of influenza infection cases and death. These observations are in line with research on previous influenza seasons and pandemics [33]. A modeling study highlighted that Sub-Saharan Africa experienced the highest annual mortality rate due to influenza (2.8–16.5 per 100,000 persons), surpassing other regions, with Southeast Asia following (3.5–9.2 per 100,000 persons) [1]. Similarly, a review of influenza hospitalization rates globally showed the highest rates in Asia and North America [44]. The substantial disease burden in Asia and Sub-Saharan Africa may stem from factors such as inadequate nutrition, limited healthcare access (including vaccinations), and other poverty-related factors that are prevalent in these regions, all of which exacerbate the risk for poor outcomes [45]. However, our findings diverge from a prior study suggesting that Africa had a lower incidence of influenza cases compared to other regions [33]. This discrepancy could be attributed to the fact that our review encompasses various types of study designs and data sources beyond 2018, in contrast to the earlier study which relied solely on data from the WHO surveillance database from 2011 to 2018.

Our analysis indicates higher incidence rates of influenza-related hospitalizations and ICU admissions among adults compared to children and adolescents in Asia and Latin America. This aligns with findings from past influenza seasons and pandemics. A study during the 2009 H1N1 pandemic in Taiwan showed a greater risk of experiencing complications requiring intensive care and higher mortality rate among adults than children [46]. Similarly, a decade-long study (2009–2019) in the United States found higher rates of hospitalization and ICU admissions among adults, especially the elderly, compared to those under 18 years old [47]. These trends are supported by epidemiological studies from various high- and low-income countries, confirming older age is a significant risk factor for severe influenza outcomes like hospitalization, ICU admission, and death [9,11,48].

### 4.2. Disease Burden of Influenza-Associated Sequelae

Data on acute influenza-associated sequelae were most frequently reported in the included studies, with relatively more data reported on the incidence and proportion of hospitalizations in Asia, Sub-Saharan Africa, and the Middle East and North Africa than in Latin America. Data on other outcomes and long-term influenza-associated sequelae were scarce.

When compared to other regions, Sub-Saharan Africa had the highest incidence rate of hospitalization due to influenza-associated sequelae. Data on incidence rates of acute influenza-associated sequelae in terms of caseload and death were reported from Asia only. This highlights the challenges in determining which regions have the highest burden of influenza-associated sequelae. The 2017 Global Burden of Disease Study showed that nearly one-third of all influenza-associated lower respiratory infection deaths occurred in India, China, and Russia [30]. Our review revealed a higher burden of hospitalization due to influenza-associated sequelae in the Middle East and North Africa than in Sub-Saharan Africa. Underlying metabolic syndrome and cardiovascular diseases are recognized risk factors for developing influenza-associated sequelae [49,50,51]. In the Middle East and North Africa, the overall disease burden attributable to metabolic risk factors (e.g., high systolic blood pressure and high fasting plasma glucose) has been steadily increasing over the past years from 5580 DALYs per 100,000 in 2013 to 6095 DALYs per 100,000 in 2019 [52]. A similar trend was observed in the Southeast Asia–East Asia–Oceania region, where a record-high 6714 DALYs per 100,000 was observed in 2019. In contrast, the metabolic risk burden in Sub-Saharan Africa has been relatively stable, fluctuating between 3053 and 3032 DALYs per 100,000 from 2013 to 2019. As the burden of non-communicable diseases in Sub-Saharan Africa is projected to rise in the coming years, the burden of acute and long-term influenza-associated sequelae could increase to the level currently seen in Asia, the Middle East, and North Africa [53]. Therefore, the prevention of communicable diseases, especially respiratory infections such as influenza, remains important as part of nations’ healthcare strategies.

Among studies which stratified the proportion of death and hospitalization attributable to influenza-associated acute sequelae, generally higher proportions were observed among adults than in children and adolescents. The risk of developing influenza and its complications increases with aging [54,55,56]. The recent expanding vaccination coverage among infants and pregnant women in certain countries might partly explain the lower burden of influenza-associated acute sequelae among children, who traditionally had a higher risk of experiencing complications from influenza infections [57]. According to the WHO influenza vaccination coverage dashboard, influenza vaccination coverage from 2018 to 2022 was rising among children in Southeast Asian countries (e.g., Brunei and Thailand) and among pregnant women in the Republic of Korea and Latin American countries (e.g., Columbia and Mexico) [58]. Vaccinating pregnant women has shown to reduce the risk of influenza-related hospitalization in infants [59].

### 4.3. Disease Burden of Influenza-like Illness

ILI data were very limited across all outcomes and regions. Most studies were conducted in Asia and Sub-Saharan Africa and mainly reported the proportion of ILI cases and hospitalizations.

Although studies in Asia reported higher incidence rates of ILI cases than the sole study from Sub-Saharan Africa, Sub-Saharan Africa bears most ILI burden. The region of Sub-Saharan Africa leads in the number of published studies concerning ILI-related hospitalizations, and also reports the highest proportions and the widest range of such cases. Though limited, the data on ILI-associated deaths place Sub-Saharan Africa on par with the Middle East and North Africa as having higher proportions compared to Latin America [5]. Observed variations across studies can be attributed to differences in definitions and guidelines on ILI diagnosis between countries and settings, disparities in access to healthcare resources, quality of surveillance systems, influenza vaccination rates, healthcare-seeking behavior, and the prevalence of risk factors for severe outcomes such as the elderly distribution in the population and underlying medical conditions [49,60,61].

Due to the absence of stratified data, direct comparisons of the ILI burden across different populations were not possible, but year-to-year variations in reported outcomes were observed. These annual fluctuations can be attributed to factors such as the emergence of new influenza virus strains, varying effectiveness of seasonal influenza vaccines, and shifts in population immunity over time [62,63], and additionally the impact of the COVID-19 pandemic.

### 4.4. Impact of the COVID-19 Pandemic

This review highlights the descending trend in the number of influenza-related studies since 2018, in incidence rates of influenza infection in Middle East and Latin America from 2019 to 2022, and in the burden of influenza and ILI-associated severe outcomes in emerging markets.

We observed significant data gaps since 2020 across all influenza outcomes and regions, compared to systematic reviews including studies before 2018 [5,64,65].

The COVID-19 pandemic also played a role in reducing the disease burden of influenza and ILI, which can be attributed to three factors. Firstly, the implementation of non-pharmacological public health interventions during the COVID-19 pandemic, such as mandatory mask-wearing and lockdowns, contributed to a substantial decrease in the transmission of COVID-19 as well as other infectious diseases, including influenza [66,67,68,69,70]. Secondly, in many countries, medical resources were prioritized for COVID-19 treatment, resulting in a decline in influenza testing and hospitalizations due to influenza [71]. Thirdly, studies reported a decline in routine health seeking for respiratory illness during the first two years of the COVID-19 pandemic [70,72,73].

COVID-19 is still evolving; however, it is important not to overlook a possible rebound of influenza in the post-pandemic COVID-19 period and invest in preventive measures against COVID-19 as well as against influenza.

### 4.5. Limitations

This systematic review has several limitations. Firstly, the heterogeneity of the included studies (e.g., different objectives, utilizing various study designs, methods and case definitions) resulted in a wide range of reported incidence rates and proportions of the outcomes. This poses challenges in directly comparing and consolidating findings. Secondly, the narrative data synthesis did not provide quantitative comparisons of the disease burden, and the conclusions drawn based on the observed trends of the data should be approached with caution. The review results serve as a preliminary exploration of the data, highlighting the need for future research that might include more rigorous, statistically driven analyses like meta-analyses to validate our observations. Thirdly, there is a scarcity of published data on a number of influenza- and ILI-associated outcomes, particularly in the Middle East, North Africa, and Latin America.

## 5. Conclusions

Despite limited data from low- and middle-income countries, this systematic review indicates that influenza causes significant disease burden in emerging markets. There are significant regional differences in influenza burden and temporal variations in influenza trend before and during the COVID-19 pandemic. Incidences of lab-confirmed influenza cases in Asia and Sub-Saharan Africa were higher compared to Latin America and the Middle East and North Africa. This systematic review highlights lacking data on the outcomes of sequelae cases, hospitalization, death, and outpatient visits related to influenza-associated acute sequelae in emerging market countries. Robust large real-world studies using a similar methodology are needed to have more accurate estimates and compare studies within age groups and regions. Continuous monitoring of the influenza epidemiology is important to inform vaccine programs in emerging markets with heavy influenza disease burden.

## Figures and Tables

**Figure 1 vaccines-12-01251-f001:**
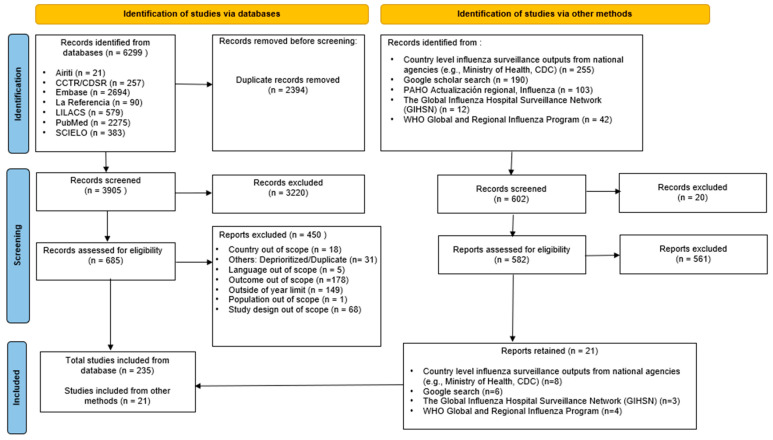
PRISMA flow diagram. Abbreviations: CDC—Centers for Disease Control and Prevention; CDSR—Cochrane Database of Systematic Reviews; CCTR—Cochrane Central Register of Controlled Trials; LILACS—Latin American and Caribbean Literature on Health Sciences; PAHO—Pan American Health Organization; SCIELO—Scientific Electronic Library Online; WHO—World Health Organization.

**Figure 2 vaccines-12-01251-f002:**
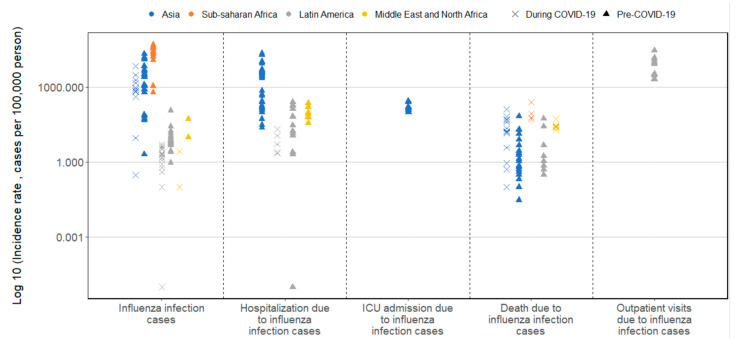
Overview of incidence rate for lab-confirmed influenza disease burden and its associated outcomes in 4 regions (n = 32 studies). This figure includes all incidence rates reported in the studies to ensure comprehensiveness of the result presentation. The wide range of incidence rates is due to the various study populations, periods, study designs, and methods used in the included studies.

**Figure 3 vaccines-12-01251-f003:**
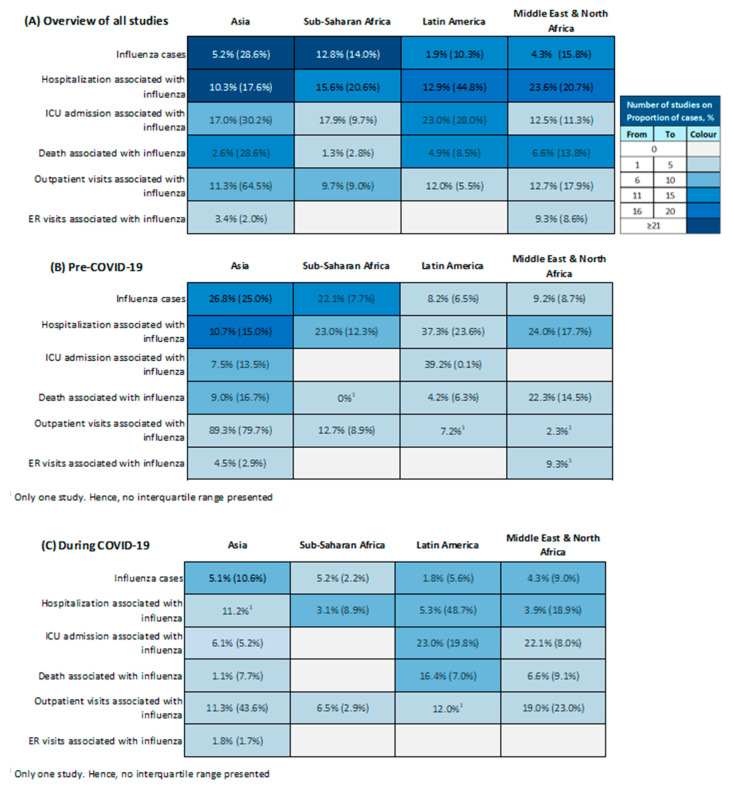
The median and interquartile range (IQR) of proportions of lab-confirmed influenza cases and its associated outcomes reported (**A**) in all eligible studies; (**B**) in studies that were conducted before the COVID-19 pandemic (2018–2019); and (**C**) in studies that were conducted during the COVID-19 pandemic (2020–2023) (n = 244 studies).

**Figure 4 vaccines-12-01251-f004:**
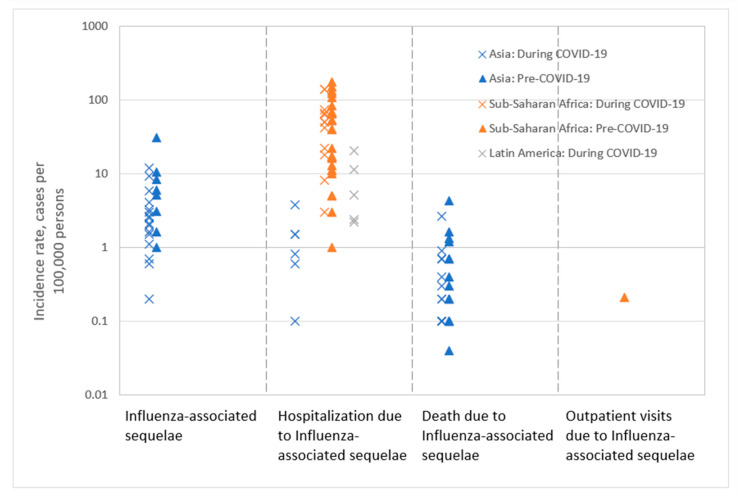
Overview of incidence rate for outcomes associated with acute sequelae due to lab-confirmed influenza infection (n = 6 studies). This figure includes all incidence rates reported in the studies to ensure comprehensiveness of the result presentation. The wide range of incidence rates is due to the various study populations, periods, study designs, and methods used in the included studies.

**Figure 5 vaccines-12-01251-f005:**
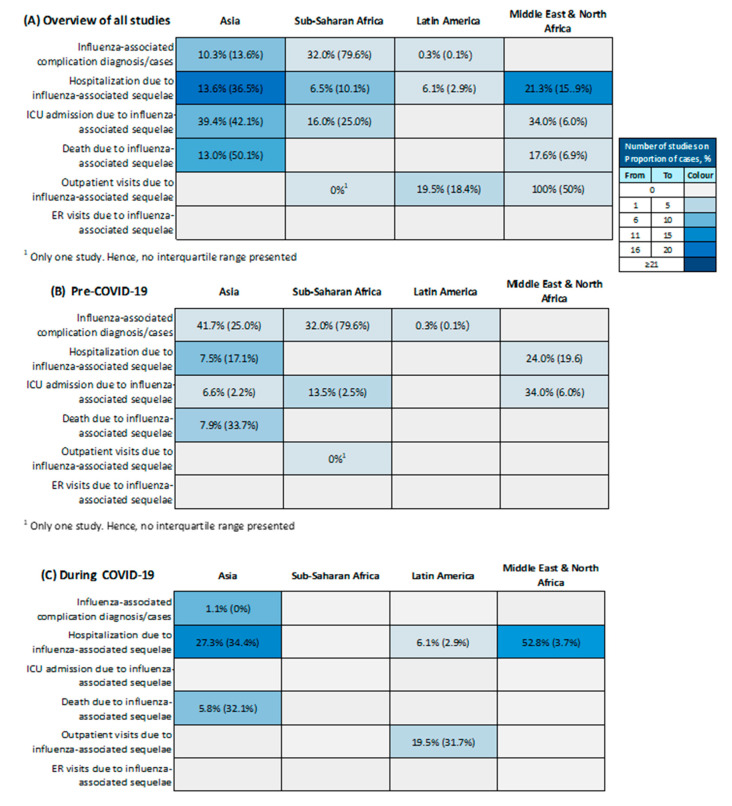
The median and interquartile range (IQR) of proportion of lab-confirmed influenza-associated acute sequelae and its associated outcomes reported (**A**) in all eligible studies, (**B**) in studies that were conducted before the COVID-19 pandemic (2018–2019), and (**C**) in studies that were conducted during the COVID-19 pandemic (2020–2023) (n = 24 studies).

**Table 1 vaccines-12-01251-t001:** Study characteristics.

Study Characteristics	Overall	Asia	Sub-Saharan Africa	Latin America	Middle East and North Africa
All, n (%) ^1^	256 (100%)	119 (46.5%)	45 (17.6%)	56 (21.9%)	47 (18.4%)
Study design, n (%) ^1,2^					
Cross-sectional study	116 (45.3%)	57 (47.9%)	17 (37.8%)	25 (44.6%)	16 (34.0%)
Cohort study	85 (33.2%)	43 (36.1%)	11 (24.4%)	17 (30.4%)	19 (40.4%)
Surveillance study	48 (18.8%)	16 (13.4%)	17 (37.8%)	11 (19.6%)	11 (23.4%)
Modeling study	1 (0.4%)	1 (0.8%)	-	-	-
Cost-effectiveness study	4 (1.6%)	1 (0.8%)	-	3 (5.4%)	-
Interventional study	2 (0.8%)	1 (0.8%)	-	-	1 (2.1%)
Population type, n (%) ^1,3^					
By age group					
Adult	40 (15.6%)	23 (19.3%)	3 (6.7%)	9 (16.1%)	5 (10.6%)
Children and adolescents	43 (16.8%)	18 (15.1%)	7 (15.6%)	8 (14.3%)	10 (21.3%)
Both adults and children and adolescents	113 (44.1%)	39 (32.8%)	26 (57.8%)	26 (46.4%)	22 (46.8%)
No age range reported	73 (28.5%)	40 (33.6%)	9 (20.0%)	14 (25.0%)	10 (21.3%)
Influenza vaccination					
Study included a vaccinated population	63 (24.6%)	13 (10.9%)	12 (26.7%)	29 (51.8%)	9 (19.1%)
% vaccinated study participants, range	0.1–98.1%	1.0–87.5%	0.1–13.6%	16.7–71.8%	0.4–98.1%
Influenza variants, n (%) ^1,4^					
Type A	178 (69.5%)	70 (58.8%)	36 (80.0%)	40 (71.4%)	32 (68.1%)
H1N1	106 (41.4%)	43 (36.1%)	17 (37.8%)	27 (48.2%)	19 (40.4%)
H3N2	78 (30.5%)	28 (23.5%)	14 (31.1%)	24 (42.9%)	12 (25.5%)
H5N1	2 (0.8%)	1 (0.8%)	-	-	1 (2.1%)
H5N8	2 (0.8%)	1 (0.8%)	-	-	1 (2.1%)
H7N9	5 (2.0%)	2 (1.7%)	1 (2.2%)	1 (1.8%)	1 (2.1%)
Not specified	53 (20.7%)	20 (16.8%)	15 (33.3%)	9 (16.1%)	9 (19.1%)
Type B	145 (56.6%)	55 (46.2%)	30 (66.7%)	32 (57.1%)	28 (59.6%)
B/Yamagata	32 (12.5%)	9 (7.6%)	4 (8.9%)	14 (25.0%)	5 (10.6%)
B/Victoria	39 (15.2%)	10 (8.4%)	6 (13.3%)	16 (28.6%)	7 (14.9%)
B/Yamagata and B/Victoria	32 (12.5%)	9 (7.6%)	4 (8.9%)	14 (25.0%)	5 (10.6%)
Not specified	105 (41.0%)	44 (37.0%)	24 (53.3%)	16 (28.6%)	21 (44.7%)
Type C	4 (1.6%)	1 (0.8%)	1 (2.2%)	1 (1.8%)	1 (2.1%)
Type D	1 (0.4%)	1 (0.8%)	-	-	-
Not reported	86 (33.6%)	48 (40.3%)	10 (22.2%)	16 (28.6%)	12 (25.5%)
Influenza category, n (%) ^1,4^					
Lab-confirmed influenza infection	218 (85.2%)	89 (74.8%)	39 (86.7%)	52 (92.9%)	38 (80.9%)
Lab-confirmed influenza-associated sequelae	72 (28.1%)	39 (32.8%)	11 (24.4%)	10 (17.9%)	12 (25.5%)
Influenza-like illness (ILI)	45 (17.6%)	17 (14.3%)	17 (37.8%)	4 (7.1%)	7 (14.9%)

^1^ Studies spanning multiple regions have been included in the counts for each applicable region category. ^2^ Percentages may not total 100% due to rounding. ^3^ Studies encompassing various types of study populations have been included in the counts for each category where applicable. ^4^ Studies investigating multiple influenza variants or categories have been accounted for in each applicable category.

## Data Availability

No new data were created or analyzed in this study. Data sharing is not applicable to this article.

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
