# Peer review of "Systematic Review on Influenza Burden in Emerging Markets in 2018–2023—An Evidence Update to Guide Influenza Vaccination Recommendations"

_vaccines, 2024, doi:10.3390/vaccines12111251_

Round 1

Reviewer 1 Report

Comments and Suggestions for Authors

The manuscript [vaccines-3257350], entitled Systematic Review on Influenza Burden in Emerging Markets 2018-2023 – An Evidence Update to Guide Influenza Vaccination Recommendations by Prof. Moe H Kyaw et al., reports novel progress in influenza-related studies in emerging markets.

This is an interesting and important study.

Some concerns are listed below for consideration in revision.

      Major concerns:

1.      Introduction section, this section need improving, many sentences displayed poor relation with the main idea of the article. Some paragraphs need reconstructing, such as the second and the third paragraph, they were illogical.

2.      M&M section, are all the datasets were Influenza type A? The influenza types should be clarified.

3.      Moderate editing of English language required. Some sentences are very difficult to understand/incomprehensible.

Some specific minor concerns:

4.      Line 152: What does “Data Synthesis” mean.

5.      Line 434, the conclusion should be rewritten to display the main topic of this research.

Comments on the Quality of English Language

Moderate editing of English language required. Some sentences are very difficult to understand/incomprehensible.

Author Response

Comments and Suggestions for Authors

The manuscript [vaccines-3257350], entitled “Systematic Review on Influenza Burden in Emerging Markets 2018-2023 – An Evidence Update to Guide Influenza Vaccination Recommendations” by Prof. Moe H Kyaw et al., reports novel progress in influenza-related studies in emerging markets.

This is an interesting and important study.

Some concerns are listed below for consideration in revision.

Major concerns:

  1. Introduction section, this section need improving, many sentences displayed poor relation with the main idea of the article. Some paragraphs need reconstructing, such as the second and the third paragraph, they were illogical.

Thank you for this comment. The order of the introduction paragraphs is changed and part of the text is rephrased.

  1. M&M section, are all the datasets were Influenza type A? The influenza types should be clarified.

Information on the influenza variant was extracted from the included articles, though not reported in one-third of the studies (see Table 1 Study characteristics). The summary tables in the supplementary material include stratified data for influenza variants and the main results on influenza variants are highlighted in the results section of the manuscript. To be more specific in the materials and methods section this data was extracted, we added the influenza variant and category to the following sentence: “Information was extracted on the influenza variant, influenza category, and burden outcomes associated with influenza, ILI, influenza sequalae and ILI sequalae (Table S3).”

  1. Moderate editing of English language required. Some sentences are very difficult to understand/incomprehensible.

The manuscript is reviewed for English language and edited where necessary.

Some specific minor concerns:

  1. Line 152: What does “Data Synthesis” mean.

Section 2.4 is a description of the data synthesis. Data synthesis in a systematic review includes synthesizing the findings of primary studies and when possible some form of statistical analysis of numerical data. The synthesis method depends on the aim of the review, included study designs and the type and availability of data.

  1. Line 434, the conclusion should be rewritten to display the main topic of this research.

The conclusion text is extended/rephrased.

Comments on the Quality of English Language

Moderate editing of English language required. Some sentences are very difficult to understand/incomprehensible.

The manuscript is reviewed for English language and edited where necessary.

Reviewer 2 Report

Comments and Suggestions for Authors

The article refers to a meta-analysis of published data on influenza infection and vaccination in 2018-2023. The rationale of the article is adequate. The methodology of the article is adequate. The design of the study is well represented in Figure 1. The results are clearly presented in Table 1, Figures 3, and 5, but in Figures 2 and 4, there is no complete information, and they are confusing. It would be important to show only the relevant information on parts 1,2 and 4 in Figure 2 and avoid the last column on Figure 4. This leads me to propose the limitations of the study, which should be included at the end. 

Author Response

Comments and Suggestions for Authors

The article refers to a meta-analysis of published data on influenza infection and vaccination in 2018-2023. The rationale of the article is adequate. The methodology of the article is adequate. The design of the study is well represented in Figure 1. The results are clearly presented in Table 1, Figures 3, and 5, but in Figures 2 and 4, there is no complete information, and they are confusing. It would be important to show only the relevant information on parts 1,2 and 4 in Figure 2 and avoid the last column on Figure 4. This leads me to propose the limitations of the study, which should be included at the end. 

The authors would like to thank the reviewer for the careful review of our manuscript. The aim of the figures is to provide an overview of the available data on influenza outcomes for the different regions. We believe it is also important to highlight the lacking information on influenza in emerging markets and prefer not to delete the columns with less information. In the manuscript this is highlighted in a footnote below the figures, e.g. below Figure 2: This figure includes all incidence rates reported in the studies to ensure comprehensiveness of the result presentation.

The limitations of the systematic review are described in section ‘4.5. Limitations’ (line 422-433 of the original submitted version of the manuscript), which highlights the heterogeneity, narrative data synthesis and scarcity of published data.

Reviewer 3 Report

Comments and Suggestions for Authors
  • This is an interesting review, that in addition to dealing with an important topic, also represents a significant effort in trying to comprehensively summarize a large and very diverse body of evidence, paving the road for future quantitative synthesis of data.
  • Lines 29-30: This is a systematic review. When stating that "Robust studies using a similar methodology are needed" - be specific and state what kind of studies, since you are probably referring to primary studies and not systematic reviews (i.e. similar methodology).
  • Introduction: It seems more logical to start of this section with information that is given in the third paragraph, i.e. to start of with a snapshot of the influenza burden. Definitely makes more sense to have that information before a paragraph that describes disease diagnostics.
  • Lines 59-61: It does not seem that the cited link provides data for this. These are rather contained in CDC's and WHO's reports and in the publication - Iuliano AD, Roguski KM, Chang HH, et al.; Global Seasonal Influenza-associated Mortality Collaborator Network. Estimates of global seasonal influenza-associated respiratory mortality: a modelling study. Lancet. 2018 Mar 31;391(10127):1285-1300. doi: 10.1016/S0140-6736(17)33293-2. Epub 2017 Dec 14. Erratum in: Lancet. 2018 Mar 31;391(10127):1262. doi: 10.1016/S0140-6736(18)30105-3. PMID: 29248255; PMCID: PMC5935243.
  • Lines 74-76: Is there a reference for this claim of "reduced focus on influenza"?
  • Introduction section should mention at least some basic information on influenza surveillance, as well as vaccination coverage rates.
  • Line 88: Consider specifying here the excluded countries when referring to Asia.
  • Methods: The Methods are comprehensively and adequately described, and along with the Supplementary material they allow reproducibility. In addition, what was actually done corresponds well with the registered protocol of this systematic review.
  • Figure 1: Are Cochrane library records stated anywhere?
  • Table 1: Check the sums of all percentages - somewhere they are over 100% somewhere under 100%.
  • Figures 2-5: What about also showing results for all observed emerging markets combined, like in overall in addition to showing separately.
  • Discussion section has a satisfactory level of detail, with appropriate overview of other studies and efforts on summing up influenza burden, with possible explanations provided both for the differences in some findings but also for the implications of the results in terms of the public health strategies of these geographical areas in terms of prevention of communicable diseases (underlying its importance too).
  • Lines 403-421: However, the impact of COVID-19 pandemic does not end here. While some of it still remains unknown, it is important not to overlook the possible "rebound" of influenza in the post-pandemic period. Thus, some implications should be underlined here, in terms of the importance of continuous education on non-pharmacological measures of prevention too, in addition to re-enforcing efforts for vaccination against influenza.
  • Lines 431-433: Is this sentence finished?
  • Lines 436-437: Study did not identify gaps in clinical burden, but gaps in data (available or published or..) on clinical burden.
  • Line 441: Again, clarify the "similar methodology", as mentioned in a previous comment.
  • Reference No. 13: The link is broken and accessed information is missing.
  • Reference No. 71: This is a preprint, not a peer-reviewed publication and it seems that there is a published paper from these authors on this topic - Zipfel, C. M., Colizza, V., & Bansal, S. (2021). The missing season: The impacts of the COVID-19 pandemic on influenza. Vaccine, 39(28), 3645–3648. https://doi.org/10.1016/j.vaccine.2021.05.049
  • Supplemental information: When providing summary ranges of burden estimates, and when these are rates, did you account for what kind of rates these are, and if they are all age-standardized did you check whether the same population was used as the standard?

Author Response

Comments and Suggestions for Authors

  • This is an interesting review, that in addition to dealing with an important topic, also represents a significant effort in trying to comprehensively summarize a large and very diverse body of evidence, paving the road for future quantitative synthesis of data.

We thank the reviewer for this positive feedback.

  • Lines 29-30: This is a systematic review. When stating that "Robust studies using a similar methodology are needed" - be specific and state what kind of studies, since you are probably referring to primary studies and not systematic reviews (i.e. similar methodology).

We added ‘large real-world’ to the studies in lines 29-30 and in section 5 Conclusions.

  • Introduction: It seems more logical to start of this section with information that is given in the third paragraph, i.e. to start of with a snapshot of the influenza burden. Definitely makes more sense to have that information before a paragraph that describes disease diagnostics.

Thank you for this suggestion. The order of the introduction paragraphs is changed.

  • Lines 59-61: It does not seem that the cited link provides data for this. These are rather contained in CDC's and WHO's reports and in the publication - Iuliano AD, Roguski KM, Chang HH, et al.; Global Seasonal Influenza-associated Mortality Collaborator Network. Estimates of global seasonal influenza-associated respiratory mortality: a modelling study. Lancet. 2018 Mar 31;391(10127):1285-1300. doi: 10.1016/S0140-6736(17)33293-2. Epub 2017 Dec 14. Erratum in: Lancet. 2018 Mar 31;391(10127):1262. doi: 10.1016/S0140-6736(18)30105-3. PMID: 29248255; PMCID: PMC5935243.

The suggested reference is inserted.

  • Lines 74-76: Is there a reference for this claim of "reduced focus on influenza"?

This trend was observed by screening the influenza literature. We deleted this claim from the introduction section. No new reference is inserted.

  • Introduction section should mention at least some basic information on influenza surveillance, as well as vaccination coverage rates.

The introduction is extended with a paragraph on influenza surveillance and vaccination rates:

“The  influenza pandemic in 2009 prioritized implementing preventive measures for influenza in public health policies. It became apparent that a standardized approach to epidemiological surveillance for influenza was needed, and in response the WHO developed global epidemiological surveillance standards for influenza. However, particularly in emerging markets influenza surveillance is still limited. Seasonal influenza vaccination is an effective preventive measure, though a recent meta-analysis showed low global influenza vaccination rates: 25% in the general population, 42% in persons with chronic diseases, 37% in healthcare workers, and 26% in pregnant women. Geographic differences were highlighted, with relatively high influenza vaccination rates in the American region, low in the European and Western Pacific regions, and even lower in the Eastern Mediterranean, South-East Asian, and African regions.”

  • Line 88: Consider specifying here the excluded countries when referring to Asia.

The excluded countries for Asia are added.

  • Methods: The Methods are comprehensively and adequately described, and along with the Supplementary material they allow reproducibility. In addition, what was actually done corresponds well with the registered protocol of this systematic review.

Thank you.

  • Figure 1: Are Cochrane library records stated anywhere?

The systematic literature search was conducted in Cochrane library including the Cochrane Central Register of Controlled Trials (CCTR) and the Cochrane Database of Systematic Reviews (CDSR), as indicated in the PRISMA flowchart. To clarify on the specific databases, the following abbreviations of the databases or literature sources have been added to figure 1:

Abbreviations: CDC - Centers for Disease Control and Prevention; CDSR - Cochrane Database of Systematic Reviews; CCTR - Cochrane Central Register of Controlled Trials; LILACS - Latin American and Caribbean Literature on Health Sciences; PAHO - Pan American Health Organization; SCIELO - Scientific Electronic Library Online; WHO - World Health Organization.

  • Table 1: Check the sums of all percentages - somewhere they are over 100% somewhere under 100%.

This is indeed correct, since 1. studies spanning multiple regions have been included in the counts for each applicable region category, 2. studies encompassing various types of study populations have been included in the counts for each category where applicable, and 3, studies investigating multiple influenza variants or categories have been accounted for in each applicable category. In addition, percentages may not total 100% due to rounding. We added a footnote regarding rounding for the percentages of the study designs.

  • Figures 2-5: What about also showing results for all observed emerging markets combined, like in overall in addition to showing separately.

To show the variability and gaps in data on the clinical burden in emerging markets, we prefer to present regional data in these figures. In addition, combining data from different regions might have limitations, caused by underlying regional differences in for example influenza epidemiology and healthcare policies (e.g. seasonal influenza vaccination).

  • Discussion section has a satisfactory level of detail, with appropriate overview of other studies and efforts on summing up influenza burden, with possible explanations provided both for the differences in some findings but also for the implications of the results in terms of the public health strategies of these geographical areas in terms of prevention of communicable diseases (underlying its importance too).

Thank you.

  • Lines 403-421: However, the impact of COVID-19 pandemic does not end here. While some of it still remains unknown, it is important not to overlook the possible "rebound" of influenza in the post-pandemic period. Thus, some implications should be underlined here, in terms of the importance of continuous education on non-pharmacological measures of prevention too, in addition to re-enforcing efforts for vaccination against influenza.

We agree with the reviewer and implemented his suggestions in this paragraph: “COVID-19 is still evolving, however it is important not to overlook a possible rebound of influenza in the post-pandemic COVID-19 period and invest in preventive measures against COVID-19 as well as against influenza.”

  • Lines 431-433: Is this sentence finished?

Yes, the sentence was finished, but since this question is raised we rephrased it: “Thirdly, there is scarcity of published data on a number of influenza- and ILI-associated outcomes, particularly in Middle East, North Africa and Latin America.”

  • Lines 436-437: Study did not identify gaps in clinical burden, but gaps in data (available or published or..) on clinical burden.

We rephrased “..this systematic review has identified data gaps in the clinical burden of influenza..” in “..this systematic review has identified gaps in data on the clinical burden of influenza..”

  • Line 441: Again, clarify the "similar methodology", as mentioned in a previous comment.

The studies in this sentence are specified as ‘large real-world’ studies based on the previous comment.

  • Reference No. 13: The link is broken and accessed information is missing.

Based on a prior reviewer comment, this reference is replaced by an alternative reference.

  • Reference No. 71: This is a preprint, not a peer-reviewed publication and it seems that there is a published paper from these authors on this topic - Zipfel, C. M., Colizza, V., & Bansal, S. (2021). The missing season: The impacts of the COVID-19 pandemic on influenza. Vaccine, 39(28), 3645–3648. https://doi.org/10.1016/j.vaccine.2021.05.049

The suggested reference is inserted.

  • Supplemental information: When providing summary ranges of burden estimates, and when these are rates, did you account for what kind of rates these are, and if they are all age-standardized did you check whether the same population was used as the standard?

We have included a diversity of studies (in terms of study design, inclusion criteria, analysis methods) to ensure a comprehensive overview of all relevant evidence. Many of the included studies were not population-based and employed straightforward designs and analyses often without reporting age-standardized rates. Consequently, we extracted and summarized the crude rates from these studies, as these were the only rates reported in the studies. But, as indicated in the study limitations, we are aware that this approach introduces a degree of variability of data and poses some challenges in directly comparing the findings.

Reviewer 4 Report

Comments and Suggestions for Authors

This paper is a systematic review of influenza cases from 2018-2023

I think the systematic review was done well. I think the paper itself is written well. My issue with the paper, and often with systematic reviews, is why are these results occurring? What is the deeper analysis of the results and what are the implications?

I don't think this is a big change to the paper, but one that is necessary to make the paper a really good one. 

Also, I would like to see more in the conclusions. More is needed in this section

Author Response

Comments and Suggestions for Authors

This paper is a systematic review of influenza cases from 2018-2023.

I think the systematic review was done well. I think the paper itself is written well. My issue with the paper, and often with systematic reviews, is why are these results occurring? What is the deeper analysis of the results and what are the implications? I don't think this is a big change to the paper, but one that is necessary to make the paper a really good one. 

The authors would like to thank the reviewer for the review of our manuscript. This systematic review was conducted to provide insight in the available data and gaps in data on the clinical burden of influenza and ILI and their associated sequelae across emerging markets. The disease burden caused by influenza in emerging markets is significant. This insight in the impact of influenza and its variability in emerging markets, can support the development of targeted and effective public health strategies such as influenza vaccine programs. The identified data gaps underline the importance of continuous monitoring of the influenza epidemiology to inform the public health strategies in emerging markets with heavy influenza disease burden.

Also, I would like to see more in the conclusions. More is needed in this section.

The conclusion text is extended/rephrased.

Round 2

Reviewer 3 Report

Comments and Suggestions for Authors

The Authors have addressed all of the comments.